# Artificial selection improves pollutant degradation by bacterial communities

Flor I. Arias-Sánchez ®[1,2] ✉, Björn Vessman[2], Alice Haym[2], Géraldine Alberti[2] & Sara Mitri ®[2,3] ✉

Artificial selection is a promising way to improve microbial community functions, but previous experiments have only shown moderate success. Here, we experimentally evaluate a new method that was inspired by genetic algorithms to artificially select small bacterial communities of known species composition based on their degradation of an industrial pollutant. Starting from 29 randomly generated four-species communities, we repeatedly grew communities for four days, selected the 10 best-degrading communities, and rearranged them into 29 new communities composed of four species of equal ratios whose species compositions resembled those of the most successful communities from the previous round. The best community after 18 such rounds of selection degraded the pollutant better than the best community in the first round. It featured member species that degrade well, species that degrade badly alone but improve community degradation, and free-rider species that did not contribute to community degradation. Most species in the evolved communities did not differ significantly from their ancestors in their phenotype, suggesting that genetic evolution plays a small role at this time scale. These experiments show that artificial selection on microbial communities can work in principle, and inform on how to improve future experiments.

Microbial communities naturally provide us with many ecosystem functions like digesting inaccessible nutrients or cleaning wastewater. Being able to design such multi-species communities from scratch to optimize ecosystem functions would be a major biotechnological breakthrough, but knowing which species to combine and how such a choice will affect ecological and evolutionary dynamics and thereby functional dynamics is a very challenging problem.

A first intuitive approach is to collect candidate species, study their capacities through genomic and phenotypic analyses and then combine them in clever ways that are likely to result in high function[1–4]. An alternative is to automate the optimization process while remaining blind to the properties of each species. This blind approach can be taken using artificial selection[5,6].

Artificial selection – also known as "directed evolution" or simply "breeding" – is a powerful approach that takes inspiration from natural selection. Not only has it revolutionized agriculture[7], but artificial selection has also been successfully applied in chemistry to optimize industrial enzymes[8,9], or in pharmacy to reduce HIV drug production costs[10]. These success stories have sparked the idea of artificially selecting microbial communities, promising to enhance human and ecosystem health, as well as many industrial applications.

In the year 2000, Swenson et al.[11,12] published two studies selecting natural microbial communities to increase plant biomass, to degrade an environmental pollutant or to alter the pH of an aquatic ecosystem. Although selected communities occasionally improved over time, they also observed improvements in some control lines and overall, community performance didn't differ significantly from the start of the experiments. Many studies have since followed, selecting for various host effects[13–18], production or consumption of chemicals[19–21] or simply for population size[22,23]. The success of these

[1]BIH Center for Regenerative Therapies (BCRT), Charité - Universitätsmedizin Berlin, Berlin, Germany. [2]Département de Microbiologie Fondamentale, Université de Lausanne, 1015 Lausanne, Switzerland. [3]Swiss Institute of Bioinformatics, 1015 Lausanne, Switzerland. ✉e-mail: flor-ines.arias-sanchez@charite.de; sara.mitri@unil.ch

experiments has also been limited[6,24], often showing inconsistent results between repeats or only a moderate increase in function.

One fundamental difficulty with artificially selecting communities is that selection is applied at the group level – rather than the individual level as with conventional breeding – but natural selection continues to act at the individual level, resulting in little control over the ecological and evolutionary dynamics occurring within each community and within each species[5,25,26]. This is because individual organisms within each group go through several generations within each selection round, with each genotype dividing at a different rate. Over time then, (i) competition between species may lead to the extinction of slower-growing species that may contribute to community function[6,27], and (ii) assuming a trade-off between function and growth, competition within species selects for cheater mutants that do not contribute to the function and sweep to fixation[27,28].

A second problem with the existing approaches lies in how "offspring" communities are generated from their "parents" at every round: parent communities are either simply diluted to make offspring (low abundant species may go extinct) or pooled together and then distributed over the offspring communities. Both approaches result in offspring communities that are very similar to one another, and do not deviate much from the communities at the start of the experiment[20,29]. The resulting lack of variability between communities gives little material for artificial selection to work on. The challenge then is to develop a selection method that favors cooperation within and between species, while maintaining between-community variability and selecting for increased function at the community level.

Here we address these fundamental problems by experimentally testing whether a novel selection approach called "disassembly selection" that was inspired by optimization algorithms from the computational sciences called genetic algorithms[30,31] could improve community performance over the length of a selection experiment. We have previously compared our approach to existing methods theoretically[29]. Here, we implemented disassembly selection experimentally to automatically explore the species composition search space: we randomly generated communities of known species composition, and then repeatedly selected the best-scoring communities, disassembled their member species and re-assembled new communities that differed slightly in their composition for the next round (Fig. 1A). This approach was expected to improve performance while maintaining between-community variability[29]. A second goal of our disassembly approach was to limit competition within communities and instead select for increased cooperativity. To achieve this and avoid aggressive species that exclude all others, we penalized communities where species extinctions occur.

We used our approach to find a community that can efficiently degrade industrial pollutants called Metal Working Fluids (MWFs), a challenge we have previously studied using a single four-species bacterial community[32]. As this original community could only degrade 44.4% of the MWF on average, we hypothesized that there would be room for improvement.

After 18 rounds of selection using a pool of 11 species isolated from MWF in which 167 combinations of four species were tested, we found a community that degraded 75.1% of the MWF on average. While this is significantly better than our original community[32], the best community in the first round, and a random control, the magnitude of the improvement was still marginal. We also separately found a species pair that performed at least as well as the top community. The lessons learned with this approach suggest that it can still be simplified, potentially leading to more meaningful improvements in performance.

## Results

### Degradation efficiency increased over 18 rounds
Briefly, we designed a community selection method where 11 species were first randomly combined into 29 communities of four species

each, with approximately equal initial abundances. We let these 29 communities grow for four days, scored them according to their degradation ability while penalizing for species extinctions, "disassembled" the top ten by selective plating, sampled viable cells of each species and used them to rebuild a new round of 29 communities that resembled the best-scoring ones. Resemblance was achieved by either rebuilding the exact same communities as in the previous round – even with the same starting population sizes for each species – or by randomly exchanging one member species in a winning community to introduce some variability and to ensure that all 11 species remained in the meta-community. We carried out this procedure 18 times, with one round per week (Methods, Fig. 1A). To test whether our selection approach could find communities that degraded better than the random species combinations at the start and that this was due to community-level selection, we included a control treatment where ten randomly-chosen communities (not based on their degradation scores) were plated and used to build communities in the next round (Methods).

In the last round, the five best-degrading communities in the selection treatment (see Fig. S1 for comparisons of $x$ best, other than five) scored higher than the top five initial communities from either treatment (round 0: 62.28% ± 4.92 vs. round 18 selection: 73.42% ± 7.38, Wilcoxon rank-sum test with continuity correction, $df = 15$, $p = 0.012$), and than those in the last round of the random treatment (round 18 random: 63.47% ± 5.49, random vs. selection $df = 9$, $p = 0.033$). In contrast, the top five from the last round of the random treatment did not degrade significantly better than the initial communities (random vs. initial $df = 15$, $p = 0.85$, Fig. 1B).

Throughout our experiment, we tested 167 different combinations of four species (141 in the selection and 156 in the random treatment, with some overlap) out of 174 possible permutations of 11 species (some species combinations were avoided in both treatments as the species were indistinguishable using selective plates, see Methods). The selection treatment tested high-performing communities more often than the random treatment, and this occurred preferentially in the later rounds of the experiment (high density of purple dots in top right corner of the selection treatment in Fig. 1C, S2, S3), showing selection for improved degradation and the maintenance of high-performing communities. Our approach also continued to explore the search space by testing many new communities at each round: in round 18 there were still communities with low degradation scores (Fig. 1C).

In an effort to estimate the ruggedness of the community "fitness landscape"[4], we next asked whether the communities with the highest degradation scores resembled each other in their species composition, by calculating the Hamming distance to the best community (number of species that one must exchange to get the same composition as the best community, Fig. 1D). At first glance, there was no obvious pattern between the similarity in community composition to the top community and degradation score. However, the best 5 communities had a distribution of Hamming distances that was significantly lower than the distribution of distances between all pairs of communities in our study (Kruskal-Wallis $H$-test, $p = 3.9 \times 10^{-4}$, Fig. S4).

### Selection reduced extinctions, but did not increase evenness or total biomass
We first explore whether selection has favored certain community properties: population growth, community evenness or species survival. When designing our selection method, we decided to penalize extinctions by scaling the degradation scores by the fraction of surviving species. We did this firstly to ensure that communities did not reduce to single species, and secondly to test whether we could select against competitive species, or even for more cooperative mutants. Extinctions were quantified in each round by plating 10 communities

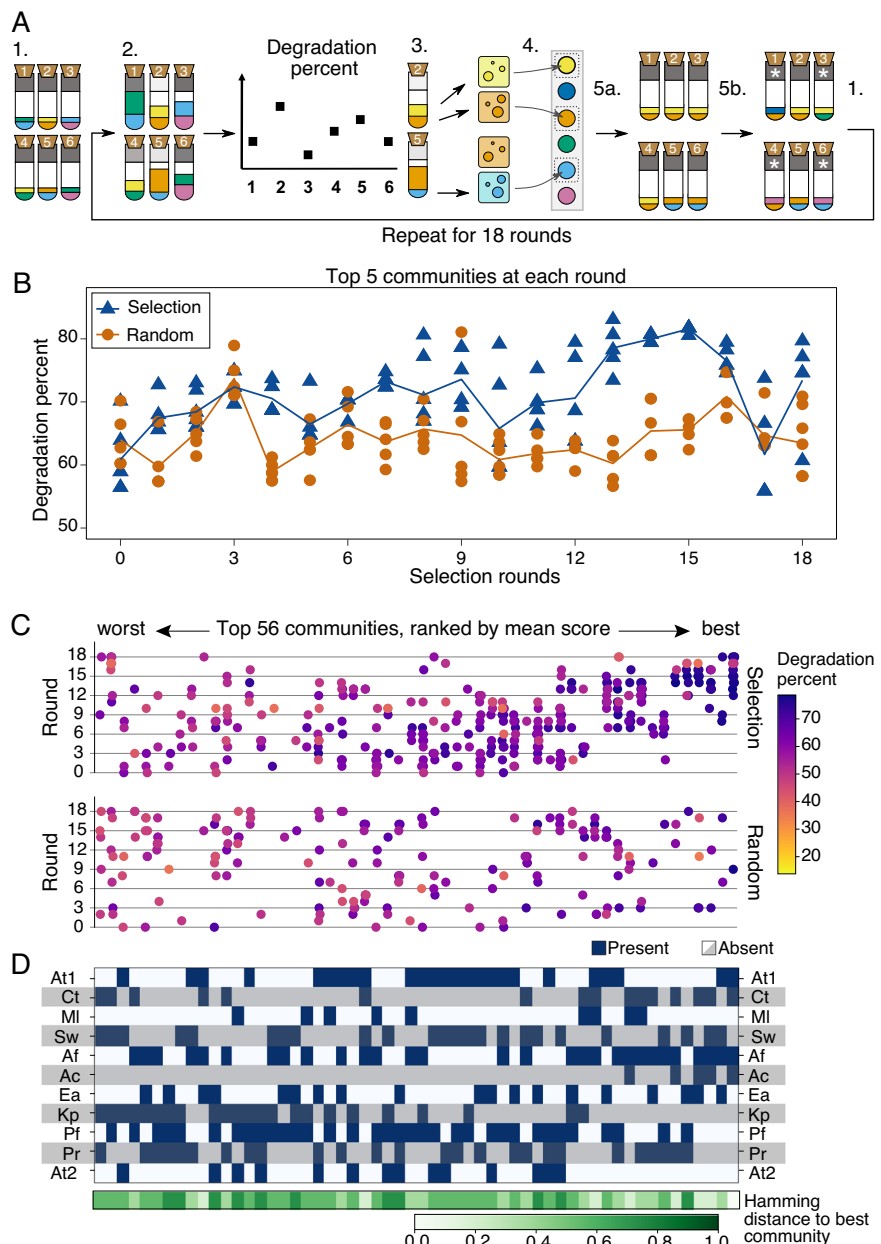

**Fig. 1 | Selection method and its performance. A** Illustration of the selection method (see Methods for details). Each tube represents a community of 4 species (2 colors drawn for illustrative purposes): 1. Define 29 communities of randomly drawn species and inoculate each community in MWF+AA. 2. Following growth, measure degradation score as the difference in pollution load to an abiotic control, illustrated by the gray field at the top of each tube. 3. Select the communities with top 10 degradation percents (illustrated by tubes 2 and 5 here) and plate these on selective media to separate their members. Plating allows to combine degradation percent with extinctions to calculate community scores. 4. Collect viable cells of each species from the corresponding community with the highest score and freeze down. 5a. Generate 29 new communities in proportion to community scores. 5b. Randomly choose 21/29 of the new communities (illustrated with 4) for species exchange. Remove one resident species at random and introduce a new species in its place. Assemble the new communities in the lab using the frozen species from the previous round and repeat from step 2. **B** Degradation scores of the 5 best communities in each round for the selection (blue triangles) and random (orange circles) treatments, with lines through the average of the 5. **C** Community composition (x-axis) vs. degradation score (hue, color bar) for the best 56 communities (one third of all 167 tested communities, the full set is shown in Fig. S2) over the 18 rounds of selection (y-axis) in the selection (top panel) and random (bottom panel) treatments. Communities on the x-axis are ordered by increasing degradation scores (averaged over all instances of the same species composition). Note that these are degradation percentages, not final community scores (extinctions not considered). **D** Community composition corresponding to panel C, showing the presence (dark blue) or absence (white/grey) of each species, and illustrating the difference in composition by the Hamming distance (i.e. the number of substitutions needed to transform a given community to another) to the community with the highest degradation score at the bottom. Species abbreviations are as listed in Table 1.

per treatment (see Methods) on selective media on day 4 and comparing the presence of each species to how we composed the community on day 0. The distribution of extinctions per round was significantly lower in the selection compared to the random control treatment (Kolmogorov–Smirnov test, $p = 0.013$, Fig. 2A, S5, Table S1).

As a control, we also counted the number of contamination events (any species that was present at day 4, despite not being inoculated at day 0), which we did not expect to vary significantly between the treatments. Indeed, we found no significant difference in contamination events per round between the two treatments

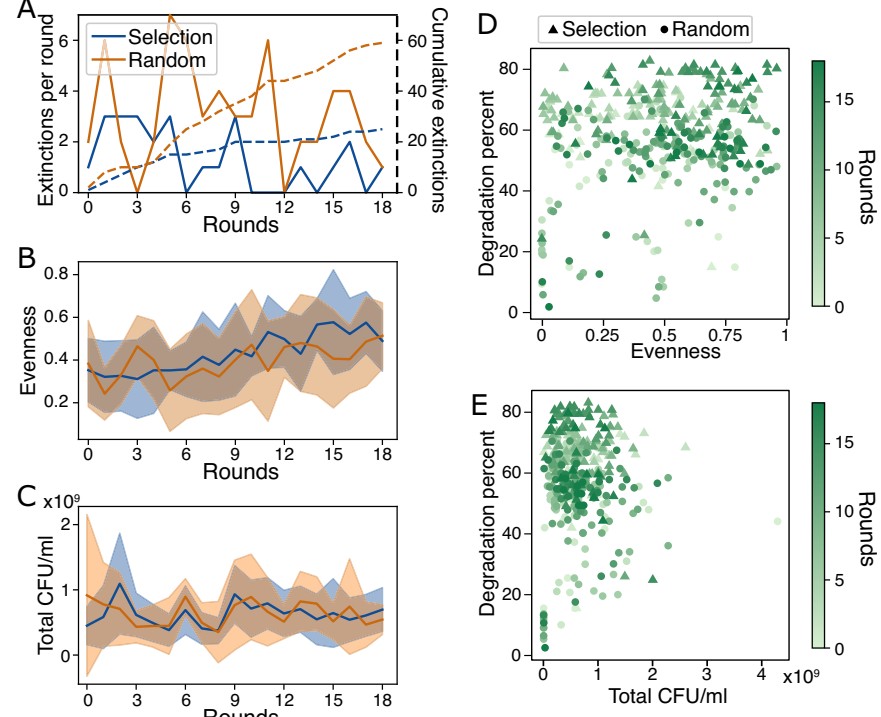

**Fig. 2 | Number of extinctions, evenness and total population size over time.**
**A** Number of extinctions per round (solid lines) and cumulative (dashed lines) in the
10 plated communities of the selection and random treatments. **B-C** Mean (lines)
± SD (shaded areas) values of the 10 plated communities at each round where (**B**)
shows evenness (the effective species number divided by its theoretical maximum
value) and (**C**) total population size in CFU/ml. **D-E** Degradation percent plotted
against (**D**) evenness and (**E**) total population size with the selection treatment in
triangles and the random treatment in circles, and color representing selection
rounds. Population size, growth or evenness could only be calculated for the 10
communities per treatment that we plated at each round (see Methods).

(Kolmogorov–Smirnov, $p = 0.8$). Despite this difference, another
explanation could be that fewer extinctions occurred in the selection
than the random treatment because selected communities more often
contained strong growers that promote the survival of others and
increase degradation score (we highlight communities lacking strong
growers in Table S1).

Next, we ask if communities in the selection treatment were more
even than in the random control. We might expect selection to favor
evenness, since species in diverse communities may complement one
another while communities dominated by a single species risk
excluding others that could contribute to degradation. Calculating
evenness as the effective species number relative to its maximum
value (Methods, Eq. (1)[33]), the evenness of the 10 communities whose popu-
lations we quantified increased with time in both treatments, but did
not differ significantly between treatments (Fig. 2B, ANCOVA (even-
ness - rounds + treatment), treatment: $p = 0.735$, rounds: $p < 0.001$). We
therefore conclude that selection did not favor more even communities
compared to the random treatment.

Finally, we might expect the total biomass in communities to
influence degradation for two reasons: (i) degradation could be the
aggregated effect of individual cells assuming that all species contribute
to degradation, and (ii) as species adapt to the medium, they might
increase their growth rates, which should increase degradation. We
calculated the total population size on day 4 per community at each
round of selection, but found no significant effect of total biomass or
selection treatment on degradation: Total biomass did not correlate
strongly with time (Spearman's $\rho = 0.04, 0.07$, for selection and random,
respectively, Fig. 2C) and was not significantly different between the two
treatments. Indeed, degradation score did not even correlate with total
biomass (Fig. 2E, Spearman's $\rho = -0.0062$, $p = 0.904$).

In sum, selection seems to have favored communities whose
members are less likely to drive each other extinct, but no other

community features could explain the increase in degradation scores
or the difference between treatments.

## Successful communities were composed of good degraders, their facilitators and freeriders

Noticing that certain species were often found in the winning com-
munities, we next explored which species features were selected and
whether community degradation scores depended on the presence of
specific species or species combinations.

First, we analyzed which species were over- or under-represented
in the meta-community compared to what one would expect by
chance. For each treatment, we quantified how often each species
appeared among plated communities in the last 5 rounds of the
experiment. If a species' frequency was more than one standard
deviation above or below the frequency one would expect by chance
(18.18), we designate it as over- or under-represented, respectively
(mean ± SD = 18.18 ± 11.8; Fig. 3A dashed line, shaded area). Over-
represented species were: Ct, Af and Ac in the selection treatment, and
Pf and Pr in the random treatment, while Ml and At2 were under-
represented in the selection treatment and Ac in the random treat-
ment. The communities that contained the over-represented species
tended to be associated with high degradation scores (Fig. 3A).

The best-scoring community in the selection treatment (At1+Ct+Af
+Ac) contained all 3 over-represented species, which partially explains
their over-representation. However, it does not answer how its member
species were contributing to the score. High degradation in these
communities could either be due to single species degrading well, or to
synergistic effects between the species. To find the answer, we grew all
11 species alone and in most pair-wise co-cultures and ranked them
from best to worst degradation. We included four of the best 4-species
communities and all eleven species grown together, as a reference
(Fig. 3B). We observed a wide variation in degradation abilities, and to

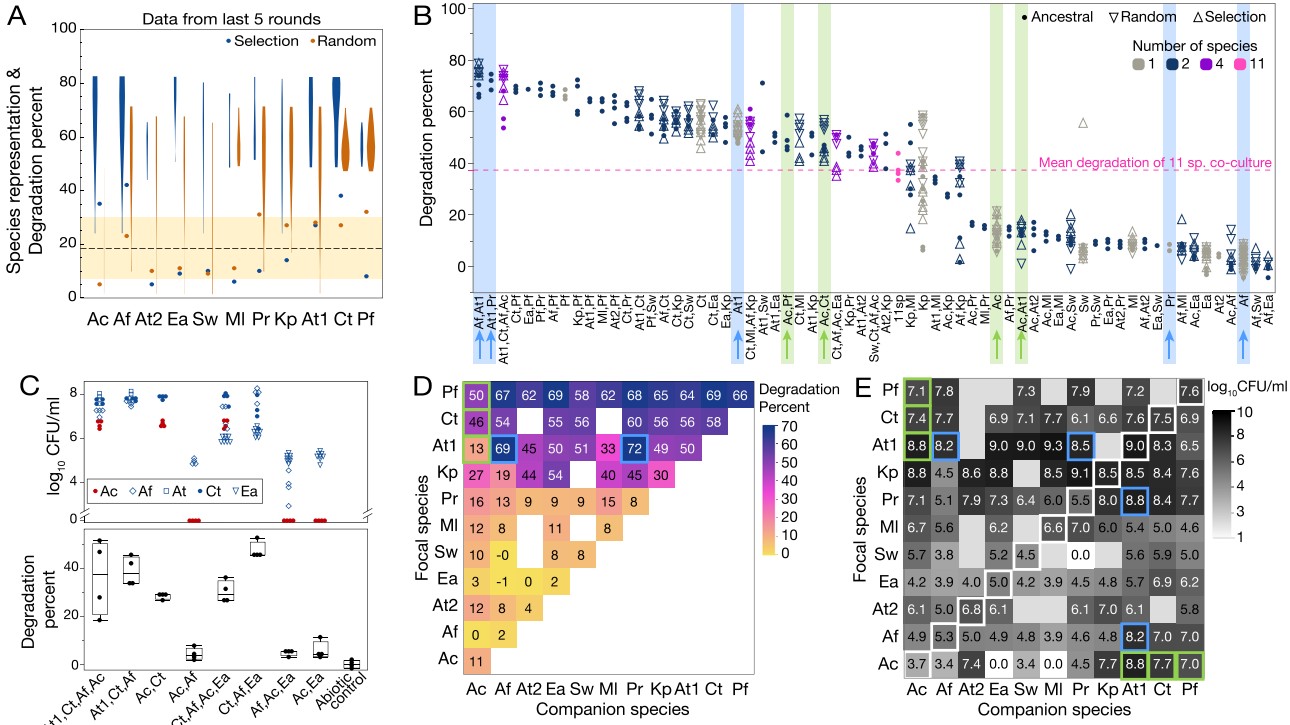

**Fig. 3 | Representation of species in evolved communities and factors that might explain it. A** Species representation and corresponding percentage of degradation in the last 5 rounds of the evolution experiment. As both measures can be quantified in percent, we display them on the same y-axis. The dashed line represents the average frequency at which we expect to see a given species in the last 5 rounds by chance, and the shaded area one standard deviation away from that average. Points that are outside the shaded area are more or less represented than expected by chance. The violin plots show the degradation scores of communities containing that species. **B** Degradation percent on day 3 in monocultures, pairwise co-cultures, top communities and 11 species together, using species taken from ancestral strains or strains isolated at the end of the random or selection treatment. Data-points are ordered according to the average degradation % and interesting cases are highlighted with a colored background and arrows

corresponding to data shown in panels (**D**) and (**E**). **C** Experiment to determine whether Ac might be a "free-rider" (in 4 technical replicates). Data points in the top panel show population sizes ($\log_{10}$CFU/ml) of different species and boxplots in the bottom panel show the distribution of degradation scores at day 3 of co-cultures as indicated on the x-axis. Each box shows the first, second and third quartiles and the whiskers the minimum and maximum values. Ac reduces the degradation score of the communities it is in, or increases their variance. **D** Matrix of degradation percentage in mono- (diagonal elements) and co-cultures of ancestral strains only (average of dots in panel (**B**)). **E** Matrix of population sizes ($\log_{10}$CFU/ml) in mono- (diagonal elements highlighted with white squares) and pairwise co-cultures of ancestral strains only. In panels (**B**), (**D**) and (**E**) we highlight interesting cases in blue and light green that are further discussed in the text. Species abbreviations are as listed in Table 1.

our surprise, all 11 species together ranked 35th (dashed line in Fig. 3B), which is well below what even single species could achieve.

The best individual degraders were Pf, Ct and At1 (mean degradation: 66%, 58% and 50% respectively), while Af, Ea and At2 were the worst (mean degradation: 2%, 2% and 4% respectively, Fig. 3B, D). Interestingly, Af which is one of the worst degraders, was present in many winning communities. This may be because when combined with At1, it achieves one of the highest degradation scores (Fig. 3B, D blue highlight). Compared to their growth in monoculture, At1 promoted the growth of Af by more than 3 logs, although Af reduced the growth of At1 (Fig. 3E blue highlight).

Not all good degraders were over-represented in the selection treatment, though. Pf and At1 each featured in only 2 of the 10 best communities (Fig. 1D), despite Pf being the best-performing species alone and featuring in 8 of the 10 best pairs (Fig. 3B, D). In contrast, Ct was present in 7 out of the 10 winning communities (Fig. 1D).

We also analyzed which species were most present when extinctions occurred, as we selected against extinctions. In the 20 communities of the selection treatment where extinctions occurred, the species most often found were At1, Pr and Pf (13, 12 and 12, respectively, Fig. S5). In 9 of the 20 communities, At1 and Pr were both present, which may explain why they do not feature together in the best 10 communities (Fig. 1D), despite being one of the best degrading pairs (Fig. 3B, D blue highlight). In contrast, only 2 extinction events occurred in the selection treatment when Ct was present. Even if At1

was often associated with extinctions, it greatly increased the growth of Af and Pr, resulting in the best-degrading pairs (Fig. 3B, D blue highlights), of which one (At1 + Af) was present in the winning community.

The third over-represented species is Ac, which on its own was one of the worst degraders (mean degradation: 11%). And although its degradation improved greatly when together with Pf and Ct (mean degradation: 50% and 46%, respectively), these degradation scores were lower than what Pf (mean degradation: 66%) and Ct (58%) could achieve alone. Interestingly, Ac's growth was also significantly promoted by the three degrader species (Fig. 3E, Ac row, light green highlight) and while it did not reduce the growth of the degraders much, it greatly reduced their capacity to degrade, particularly for At1 (mean degradation: 13% as opposed to 50% when grown alone). These results suggest that Ac may have acted as a "free-rider" species that got carried along with the best communities. We tested this idea by removing Ac from the winning community and observed a reduced variability in its performance (Fig. 3C). Removing it from the fourth-best community also significantly increased the community degradation score (degradation of Ct + Af + Ac + Ea: 30.2 ± 4.5%, vs. Ct + Af + Ea: 47.3 ± 3.5%, t-test, p = 0.0012), whereas removing Ct from this community drastically reduced its degradation score (Af + Ac + Ea: 4.7 ± 1.2%, p < 0.001, Fig. 3C).

In sum, selection appears to have favored communities with at least one good degrader species, especially if its score could be

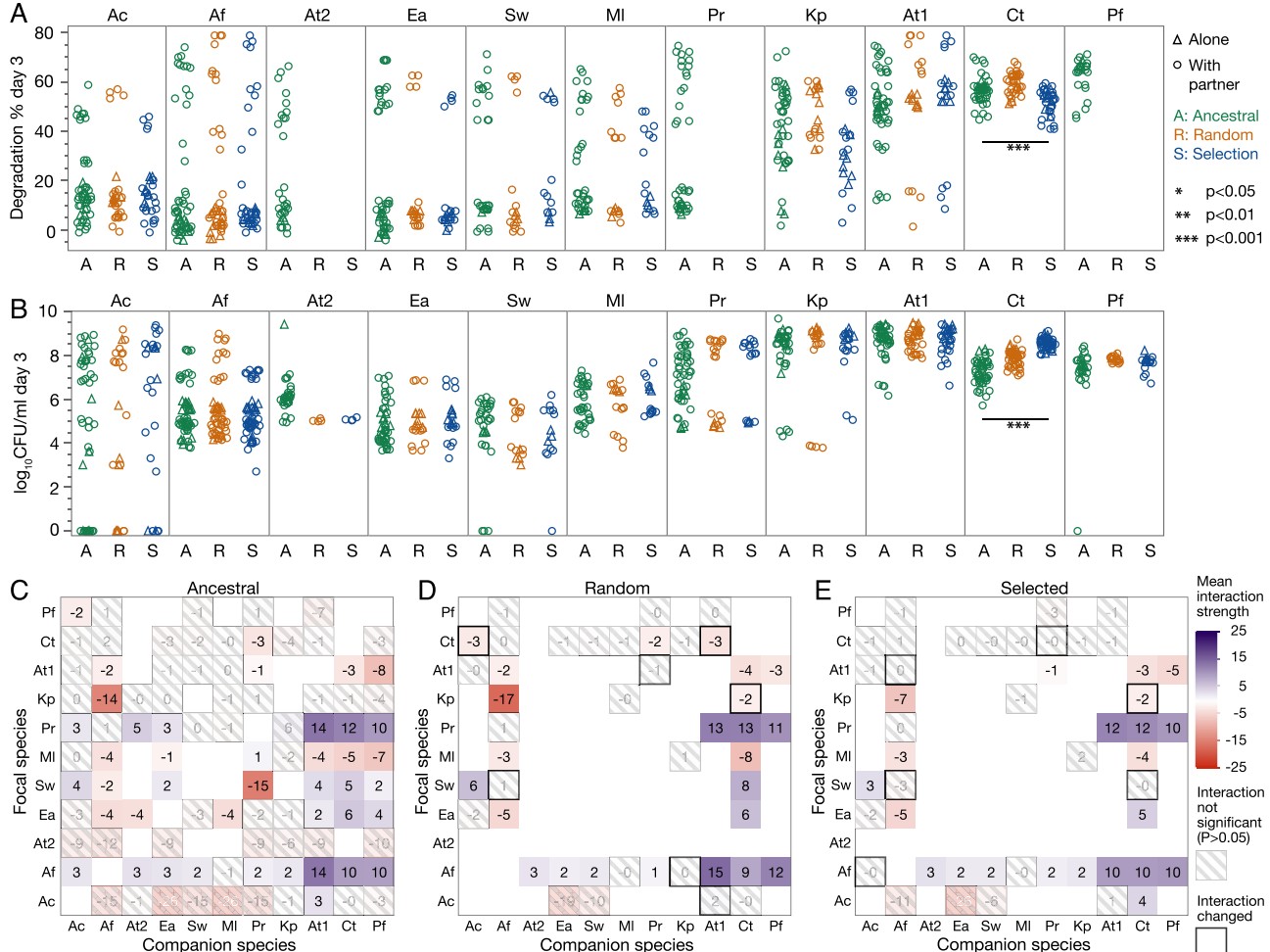

**Fig. 4 | Effect of within-species evolution. A** Degradation percent or (**B**) population size (log₁₀CFU/ml) of each species on day 3 in three conditions: the ancestral strains before the experiment, strains harvested after 18 rounds of selection treatment (S) and after 18 rounds of the random control treatment (R). Data from mono- (alone) and pairwise co-cultures are shown (with partner). Significant differences are calculated using a generalized linear model with biological replicate as random variable and number of species in culture as an explanatory variable, significant *p*-values with a Bonferroni correction for multiple comparisons are shown. Treatment only had a significant effect on Ct (detailed statistics in main text). **C**–**E** Interactions between ancestral species (**C**), the species evolved in the random

treatment (**D**) and the selection treatment (**E**) defined as the log₂ fold-change of the focal species in CFU/ml of day 3 in co-culture with the companion species vs mono-culture. Interactions that were not significant (no significant difference between growing alone or with companion species) are shaded, *p*-values were adjusted using the Benjamini-Hochberg method. Positive (facilitative) interactions are in blue, while negative interactions are shown in red. White squares are ones that we did not measure. Overall, we saw very few changes between ancestral and evolved species (black-bordered squares, quantified in Table S2). Species abbreviations are as listed in Table 1.

enhanced by "weaker" species, as long as they did not cause extinctions. This approach does not seem to eliminate free-riders that appear in the final communities despite their deleterious effects on degradation scores.

**Did the best communities improve compared to their ancestors?**

Up to this point, we have viewed our disassembly selection approach as a way to recombine different species to increase degradation scores, but have not considered whether any of the species evolved. To determine whether the species at the end of the experiment differed phenotypically from their ancestors, we compared the degradation and population sizes after 3 days of all 11 species isolated from the different treatments to their ancestors (Fig. 4A, B). We observed no significant differences for any species except Ct. Ct isolated from the selection treatment grew significantly better than its ancestor $(3.8 \times 10^8 \pm 2.2 \times 10^8$ vs. $3.3 \times 10^7 \pm 4.1 \times 10^7$, Wilcoxon rank sum, *df* = 79, *p* = 4.72 × 10⁻¹⁶) and than its counterpart isolated from the random control treatment $(9.92 \times 10^7 \pm 8.68 \times 10^7$, *df* = 71, *p* = 1.1 × 10⁻¹²). The

strain of Ct from the selection treatment also degraded significantly worse than its ancestor (51.7 ± 5.2 vs. 56.4 ± 5.5%, *df* = 79, *p* = 0.00054), suggesting that it may have evolved to invest more into biomass and less into degradation, but we do not explore this idea further.

We were also curious whether inter-species interactions had changed over time. Indeed, we had chosen to conduct this experiment in growth medium containing casamino acids (see Methods), as we expected from previous work for competition to be stronger in this environment compared to MWF without casamino acids[32] and we wondered whether selection could reduce competition.

We first used the population size data of mono- and co-cultures (Fig. 3E) to estimate interactions in the ancestral species (Fig. 4C). We then selected isolates of a few pairs (we favored species whose ancestral interactions were significant and members of the winning communities) from the random and selection treatments for which we conducted mono- and co-cultures to estimate the interactions between the evolved species. While some interactions differed after evolution (Fig. 4D, E), we found little evidence of reduced competition in the selection treatment, and more generally, no overall pattern

(Table S2). We therefore conclude that evolution at this timescale has not had profound effects on the species' phenotypes.

## Discussion

Previous community selection experiments have struggled to show consistent improvements in community functions compared to controls[6,24]. We have devised and tested a selection method to improve the degradation of MWF pollutants in small synthetic bacterial communities. The disassembly method automatically searches for species combinations with high degradation scores, while selecting against species that cause the extinction of other community members. The best community found using this approach performed significantly better than the best initial communities (Wilcoxon rank sum test, $p < 0.001$) and 69% better than the community studied in our previous work (Fig. S6, Wilcoxon rank sum test, $p = 0.007$)[32].

While this study did not directly compare our artificial community selection method to previous approaches, we can estimate their performance *a posteriori*. The most common approach, "propagule selection", propagates the best communities in each round through dilution. To simulate propagule selection, we can imagine that the best community at round 0 (At1, Sw, Ea, Pr) would have been selected and propagated further and ask whether it would have improved. The degradation scores of (At1, Sw, Ea, Pr) throughout the experiment were significantly lower than our winning community (At1, Ct, Af, Ac) ($59.57 \pm 16.83$ versus $75.45 \pm 8.43$, Wilcoxon rank-sum test, $df = 30$, $p = 0.0002$, Fig. S7), suggesting that propagule selection is unlikely to have outperformed our approach. The second common approach, "migrant pool selection" combines the best communities and dilutes them over subsequent rounds. A proxy for this experiment is culturing all 11 species together, which is where migrant pool is eventually likely to converge. The dashed line in Fig. 3B shows that this community performs particularly poorly.

Further investigating some of the top communities and many species pairs revealed that successful degradation could be achieved by combining strong degraders with other species that might not be able to survive alone, but enhanced the degradation score when paired with the strong degrader. This simple heuristic revealed that the best overall performance was achieved by two species in co-culture: At1 and Af. Adding two more species to this pair (Ct and particularly the free-rider Ac) increased the variance in community performance, and combining all 11 species performed poorly (Fig. 3). It appears then that our optimal community of four species is in fact too rich.

Another key observation is that despite our efforts to favor within-species evolution – we sampled many colonies when disassembling communities through plating to include sufficient within-species diversity, used a competition-promoting medium, and penalized extinctions to give room for interactions to evolve to become less negative or more positive – it did not have a large effect on final population sizes, degradation abilities or inter-species interactions. One explanation may be that species are changing their biotic environment too often for selection to favor any particular interactions. In agreement with this, the only change we observed is that Ct evolved in the selection treatment grew better than its ancestor alone (Fig. 4A). It could be that Ct evolved to include cheater genotypes that invest less into degradation and more into growth, although we currently have no evidence to back this up. The up-side of finding only minor changes is that one may not need to be too concerned that species will evolve to become more competitive or invest less into community function, at least on this time-scale.

Given what we have learned, would we now perform artificial selection differently? After all, given that we measured the degradation of 1044 co-cultures in this study, we could have instead carried out six replicates of each of the 174 possible 4-species combinations. Of course, with a larger species pool, testing all possible combinations would be much more challenging. Furthermore, the improvement

achieved by this proof-of-concept study was significant yet marginal, and before we can apply our method more widely, it would need to improve performance by several folds. In addition, the approach was quite cumbersome and would not be easy to set up for a new problem.

A first question is whether the artificial selection approach is useful at all, or whether we could have predicted the composition of the best community using fewer culture experiments. To explore this question, we performed an additional analysis using a simple linear model that predicts the degradation score based on species presence/absence[4]. Including the data from all our experiments, the linear model had a reasonable fit ($R^2 = 0.8$) and would have chosen a community that performed relatively well (degradation score: 69.2%, compared to 75.1% with our method). However, if we only used mono- and co-culture data to train the model, performance dropped when we tested the model on the remaining data ($R^2 = 0.26$) and the best predicted communities ranged in performance from 40.3% – 83.1% (Fig. 5A, B). Finally, if we trained the model on measurements from the communities and then tested it on the mono- and co-culture data, its predictive ability still remained low (Fig. 5C, $R^2 = 0.32$). It would be interesting to determine the minimal amount of data needed to achieve a good prediction and explore whether other prediction methods would perform better (e.g.[34]). Overall, though, this analysis suggests that this approach would not have easily identified the best-performing community.

A second lesson could be to focus on the strength of our method as a search algorithm to efficiently explore the space of possible species combinations. Our approach is analogous to a genetic algorithm[30,31] in that solutions are encoded in a "genome" (here the species composition of each community) whose "phenotype" is measured (here degradation score plus extinctions) that is subject to "mutations" (here the exchange of a species for the next round). In addition, disassembling and reassembling communities allowed each species to evolve over the course of the experiment. But since species' phenotypes changed little over the course of the experiment (Fig. 4), it would be simpler to start communities at every round from frozen stocks of ancestral cultures and avoid the challenging experimental step of disassembling communities at each round of selection (as illustrated in Fig. 5D). We would then no longer need selective media and it would suffice to know whether species went extinct, which could be achieved through amplicon sequencing. Removing the constraint of designing selective media would allow us to greatly expand the species pool, and we would no longer need to avoid any species pairs, which limited the species combinations in this study. Moreover, by including more species and generating more combinations thereof, we expect the overall performance of the method to improve[29]. Another important modification would be to allow community size to change, as opposed to restricting it to four species as we have done here. This would involve removing or adding species independently, allowing communities to grow or shrink in size. This decoupling would increase the search space of possible combinations, but might find better solutions, for example by avoiding free-rider species like Ac to establish in so many communities.

Allowing community size to change automatically would also answer an important question for community function: how many species are actually needed to solve the problem of interest? In our previous work[32], a mathematical model predicted that in harsher environments, more species are needed to achieve maximal community function compared to permissive environments. Experimentally, degradation saturated at two species in the more permissive MWF with casamino acids, compared to three species in MWF alone[32], which is consistent with our best solution here having only two species. In hindsight, a more challenging environment might have shown a stronger improvement over the experiment and required a larger optimal community.

A final important limitation of our approach is that we fixed the initial population size of all species at each round, in order to select against cheater strains that grow quickly without contributing, and to

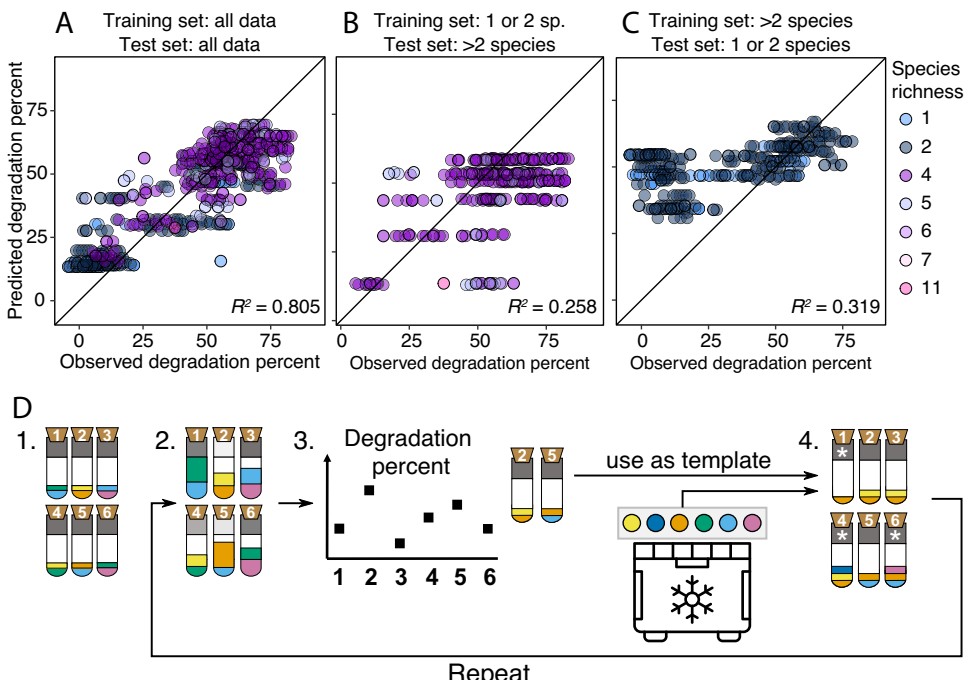

**Fig. 5 | Alternatives to our proposed method. A–C** Linear model analysis. We use a linear model (code taken from[4]) that uses species presence/absence to predict degradation percent by using (**A**) all the data we generated as training and test sets, or (**B**) the mono- and pairwise co-culture data as training set and the rest of the data as test set, or (**C**) the larger communities as training set and the mono- and pair-wise co-culture data as test set. Community richness is shown in color. Each dot is one degradation score measurement, such that biological replicates and technical replicates, if available, are all represented. $R^2$ shows Pearson's correlation coefficient. **D** Proposing a new artificial selection method. Rather than disassembling communities, we propose to use the winning communities as templates to generate the offspring communities in the next round. These communities would then be seeded by taking the clonal ancestral species from the freezer, such that there would be no within-species evolution over rounds. Step 3 would be to select the top 10 communities, 4a to generate communities in proportion to their community scores and 4b to randomly choose 21/29 of the new communities (illustrated with 4) for either species removal or introduction (see white asterisks in step 4, 4a and 4b are shown in one step). Freezer icon created by SAM Designs from Noun Project.

improve heritability of the community function[27]. It would be interesting to see how our best-performing communities would equilibrate over a few rounds of growth and dilution, when we do not adjust the initial population sizes. It is conceivable that the performance of the community at equilibrium would be different. Ideally, community stability should be part of the community score, although this would require substantial revision to the selection algorithm.

In summary, we have tested an approach to artificially select amongst communities composed of different combinations of culturable species. Our approach found a four-species community that is efficient at degrading MWF pollutants and is superior to the performance of all species in our pool grown together. However, the selection experiment was relatively complex and a smaller community was also found by testing species pairs and comparing them to the winning community. Going forward, we propose a simpler, more effective approach (Fig. 5D). Even though the challenges of ensuring ecological and evolutionary stability remain open, we argue that this first proof-of-concept study supports the blind approach to automate the breeding of bacterial communities with optimal functions.

## Methods
### Bacterial species and culture conditions
We used 11 bacterial species listed in Table 1, which were selected from a pool of 20 natural isolates from MWF (provided by collaborators) based on our ability to distinguish them on selective media. At1, Ct and Ml were previously isolated from MWF[32,35,36]. At1 was previously tagged with GFP to allow us to distinguish it on agar plates. Note that Ml (*Microbacterium liquefaciens*) was previously referred to as *Microbacterium saperdae* but a more recent classification has led us to refer

to it differently. At2 was kindly donated by Justine Collier (plant associated) and the remaining species were isolated from MWF and kindly donated to us by Peter Küenzi from Blaser Swisslube AG, a company that produces MWFs. The species were identified at Blaser Swisslube AG by MALDI-TOF, and confirmed by PCR amplification and 16S gene sequencing. All experiments were performed in 6ml batch cultures containing 0.5% (v/v) Castrol Hysol™ XF MWF (acquired in 2016) diluted in water with added salts, metal traces (Tables 2, 3), and supplemented with 1% Casamino Acids (Difco, UK). Cultures were incubated at 28 °C, shaken at 200 rpm.

**Table 1 | Bacterial species used in the experiment and the acronyms we use to we refer to them throughout the manuscript**

| Species | Our acronym |
| --- | --- |
| *Staphylococcus warneri* | Sw |
| *Agrobacterium tumefaciens* MWF001 | At1 |
| *Comamonas testosteroni* MWF001 | Ct |
| *Microbacterium liquefaciens* MWF001 | Ml |
| *Alcaligenes faecalis* | Af |
| *Aeromonas caviae* | Ac |
| *Enterococcus avium* | Ea |
| *Klebsiella pneumoniae* | Kp |
| *Pseudomonas fulva* | Pf |
| *Providencia rettgeri* | Pr |
| *Agrobacterium tumefaciens* C58 | At2 |

**Table 2 | Phosphate solution (1%) for the MWF+AA medium as described in Table 3**

| Compound | Amount |
|---|---|
| $H_2O$ | 1000 ml |
| $K_2HPO_4$ | 6 g |
| $KH_2PO_4$ | 6 g |

**Table 3 | For 600 ml of MWF+AA medium, mix in the above order, top to bottom**

| Compound | Amount |
|---|---|
| $H_2O$ | 405 ml |
| Phosphate solution | 60 ml |
| NaCl 1% solution | 60 ml |
| Casamino 1% acids solution | 60 ml |
| Hutner's vitamin-free mineral base | 12 ml |
| Castrol Hysol 100% | 3 ml |

The phosphate solution is found in Table 2. The MWF needs to be added carefully, one drop at a time to allow mixing.

## Selective media

We designed 10 selective media that allow the growth of only one or two of the 11 species at a time. Some species combinations (Ct & At2, Af & Ct, Ml & Sw, Ml & Ea, Ac & Pf, Ct & Kp) cannot be easily distinguished on these media, and we avoided combining these species in the communities of either treatment (Fig. S8A). This means that instead of the 330 combinations of 4 species out of 11, we have 174 possible communities, meaning that we are exploring a subset of the possible search space. Our selective media are generally composed of a rich base and at least one antibiotic (details in Tables S3 and S4). The disassembly plates consist of two 24-well plates where we poured 1.5 ml of each selective media into 4 wells (as shown in the 24-well templates in Fig. S8B). Because temperature was helpful to distinguish some species, we incubated some media at 28 °C and others at 37 °C. Disassembly was achieved by plating droplets of each diluted community on all the selective media (more details below). For each round, we prepared the disassembly plates one week in advance and stored them at 4 °C in the dark until they were used. Every week, the selectivity of the media was verified by inoculating 10 $\mu$l droplets from a dilution series of 2 day old cultures of all 11 ancestral species in square plates of all selective media.

## Artificial Selection

Each round of the selection experiment lasted one week and consisted of five steps (Fig. 1A): (1) assembling communities and letting them grow, (2) measuring pollution load, (3) selecting top communities and disassembling them on agar plates, (4) freezing down species samples, and (5) generating species compositions for the next round.

**Step 1: Community assembly.** In each round, we used 60 10 ml glass tubes, 29 were assigned to communities of the selection treatment, 29 to the random treatment and two tubes were abiotic controls. The number 29 was chosen because we had two racks that fit 30 tubes each; with two control tubes, each treatment was left with 29 tubes containing bacteria. The first round started with the same 29 randomly generated communities of 4 species each in the two treatments. These were drawn such that all 11 species were present in at least one community and such that species that we cannot separate with selective plates never appeared in the same community.

Communities for the first round were assembled as follows: Single colonies of each of the 11 species were picked and grown overnight in 5 mL of TSB at 28 °C, shaken at 200 rpm. The next day, cultures were adjusted to an $OD_{600}$ of 0.05 in 10 ml of PBS in a 15 ml falcon tube. For subsequent rounds, similar 15 ml tubes containing each of the 11 species for each treatment at $OD_{600} = 0.05$ were taken from the freezer (see below) and thawed. The cells of each species were then washed by centrifuging at 3220 rcf for 15 minutes and resuspended in 10 ml of MWF+AA medium (see above). For each community culture in the experiment (29 for each treatment) and the abiotic controls, 6 ml of MWF+AA were prepared in the 10ml glass tubes and 100 $\mu$l of each species were added, yielding a total of 400 $\mu$l of four species of similar relative abundances. All 60 tubes were then incubated at 28 °C and shaken at 200 rpm for four days.

**Step 2: Measuring degradation scores.** On day 4, as a proxy for pollution load, we measured the chemical oxygen demand (COD) using NANOCOLOR COD tube tests (detection range 1-15 g/l by Macherey-Nagel (ref: 985 038), see[32] for more details). We used these measurements to calculate degradation scores as $(1 - COD_4(\text{sample})/COD_4(\text{control})) \times 100$, i.e. the COD of the community after 4 days relative to the COD of the abiotic control after 4 days, in percent. Data shown in Fig. 3C was generated using expired COD tubes, which might explain why their values are different from those of the other experiments. However, given that the important comparison is between treatments within that experiment, we decided not to repeat it.

**Step 3: Selecting and disassembling top communities.** We selected the 10 out of 29 communities with the highest degradation scores from the selection treatment and 10 out of 29 communities at random from the random treatment. To disassemble the communities and determine species' population sizes, we plated dilutions ($10^{-1}$, $10^{-2}$, $10^{-4}$ and $10^{-6}$) of each community onto all selective media (see above, Fig. S8), incubated the selective plates for two days (either at 28 °C or 37 °C), and counted colony-forming units (CFUs) for each species. This allowed us to disassemble all community members, estimate population sizes and identify extinction and contamination events (species that were inoculated on day 0 but did not appear on their selective media, and species that were not inoculated in a given community but grew on selective media, respectively). We penalized extinction by scaling the degradation score of each community by the fraction of surviving species (contaminants are not counted) $0 < f < 1$ (e.g. $f = 0.5$ if only two of the four inoculated species are detected). The final community score was then calculated as $(1 - COD_4(\text{sample})/COD_4(\text{control})) \times 100 \times f$.

**Step 4: Freezing down species.** At every round of selection, we froze down a representative of each species by isolating it from the highest-scoring community where that species was present. We sampled several CFUs from the highest dilution in the relevant selective plate by adding PBS to the selected well and re-suspending by pipetting. We then adjusted the $OD_{600}$ of the samples to 0.05 in a total volume of 10 ml of PBS with 25% glycerol, then aliquoted $2 \times 1$ ml for long-term storage in cryo tubes, 3 ml for use in the following round and 5ml as backup in 15 ml falcon tubes and froze all samples at $-80$ °C. If a species went extinct in a round of selection, we recovered it from its frozen stock collected in the closest previous round.

**Step 5: Generating new species compositions.** For the following round of selection, we used a script that calculates a probability distribution from the community scores of the 10 disassembled communities and generates offspring communities by randomly sampling 29 times with replacement in proportion to this distribution. Communities with higher scores are more likely to be selected. In the random control, we sampled 29 times with uniform probability from the 10 disassembled communities.

To introduce variability into these newly generated communities, out of the 29 generated communities in each treatment, we randomly chose 21 to receive an invader species that replaced one of the four members. Both the invader and the species to be removed were chosen by uniform probability, with a few exceptions: We first chose as invaders species those that were not yet represented in any offspring communities, adding them to random receiving communities; once all 11 species were represented at least once in the new communities, we chose the remaining invaders at random but avoided invading species that were already present in the receiving community, and species that are indistinguishable from resident species on our selective media. Selection and invasion thereby result in $2 \times 29$ lists of four species each, sampled in proportion to degradation scores (or not for the random treatment) and with 21/29 of them having exchanged an old community member for a new one. We then assembled the communities in the lab from the frozen species record as described above. The script used to automatically generate offspring communities is written in python 3[37] and can be found at https://doi.org/10.5281/zenodo.12785807[38].

## Comparing ancestral and evolved strains

Following the artificial selection experiment, we conducted follow-up experiments to better understand why the selection algorithm favored certain species combinations. For each species, the frozen stocks from round 18 of the selection and random treatments were plated and incubated. Single colonies were picked and grown overnight in 5 mL of TSB at 28 °C, shaken at 200 rpm. The next day, cultures were adjusted to an $OD_{600}$ of 0.05 in 10 ml of TSB and grown for a further 3 h. The cells were then washed at 3220 rcf for 15 minutes and resuspended in 10 ml of MWF+AA medium. For each culture, 6ml of MWF+AA were prepared in 10 ml glass tubes and $100 \mu l$ of each species were added. These cultures were incubated at 28 °C, shaken at 200rpm for 3 days. CFUs were measured through serial dilution and plating on days 0, 1, 2 and 3 using the appropriate selective media (Fig. S8). We measured CODs of an abiotic control culture at day 0 and 3, and the culture tubes at day 3. The degradation scores were calculated as before. To estimate interactions between species, we grew each strain alone or with a given partner strain and compared the population size of each focal strain as the $\log_2$ fold-change in CFU/ml on day 3 in the presence or absence of the partner species. CFU/ml were quantified on selective media (Fig. S8), but on round agar plates considering all dilutions, giving higher resolution compared to the selection experiment. We used LB agar for Af and Pf instead of their selective media, as requirements were less stringent (we only needed to count them, not disassemble them) and there appeared to be differences in growth between the ancestral and evolved strains on the selective media for those species. These experiments were performed by two different authors (GA and BV), which is accounted for in the statistical analysis (see below).

## Data analysis

We used the Hamming distance between two communities to quantify the difference in species composition between them (Fig. 1). The community is in this case represented by the presence and absence of each of the 11 species, and the Hamming distance is the fraction of species mismatches. We used the implementation from the SciPy library in python[39].

We calculated evenness as the effective species number, or Hill number of order 1[40]:

$$^1D = \exp\left(-\sum_k p_k \log(p_k)\right), \qquad (1)$$

divided by its maximum value (similar to Pielou's evenness[33]), where $p_k$ is the relative abundance of species $k$ in the community. Ordinary least squares regression between evenness and round was calculated using the python package *statsmodels*[41].

We used parametric and non-parametric tests for significant differences between groups, preferring the Student's t-test for the former and the Wilcoxon rank sum test for the latter, and compared distributions using the Kolmogorov–Smirnov test. We measured correlations using Spearman's $\rho$ and quantified regressions using the ordinary least-squares implementation in the python library *statsmodels*[41]. When relevant, we corrected for multiple comparisons using the Bonferroni or the Benjamini-Hochberg method.

To compare the growth and interactions of evolved and ancestral strains, we took into account that experiments were performed by two different people. To calculate statistical significant differences in growth or degradation, *experimentalist* was taken to be a random factor in a generalized linear model (glm package in R version 4.2.2 with default parameters). To calculate interactions (one species growing in mono-versus pairwise co-culture), we only used data collected by the same experimentalist. If it was not available (only one person had measured the mono-cultures), we used this instead (see dataset 1). P-values were corrected for mutliple comparisons using the Benjamini-Hochberg method.

## Reporting summary

Further information on research design is available in the Nature Portfolio Reporting Summary linked to this article.

## Data availability

The raw data generated in this study have been deposited in the Zenodo database under https://doi.org/10.5281/zenodo.12784769.

## Code availability

All code associated with this manuscript can be found on Zenodo at https://doi.org/10.5281/zenodo.12785807[38].

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

## Acknowledgements

We thank two anonymous reviewers, Laurent Keller and members of the Mitri lab, especially Margaret Vogel and Afra Salazar for detailed comments that have greatly improved the manuscript. We thank Peter Küenzi (Blaser Swisslube) and Justine Collier (UNIL) for bacterial strains. We sincerely thank Samuele E. A. Testa, who trained BV in the lab and helped with experiments. We thank Marc Garcia-Garcerà and Bastien Vallat for helping to confirm the identity of the strains from Blaser. FAS, BV, GA, AH and SM were funded by H2020 European Research Council grant 715097. SM was additionally funded by the Swiss National Science Foundation Eccellenza grant PCEGP3_181272 and the National Center of Competence in Research Microbiomes grant SNF 51NF40_180575. FAS was additionally funded by the Swiss National Science Foundation Early Postdoc Mobility grant SNSF 181582 and the German Science Foundation Principal Investigator grant DFG AR1359/1-1.

## Author contributions

F.A.S., S.M. and B.V. conceived the project. F.A.S. developed the experimental methods. B.V. implemented the script used in the selection experiment. A.H. performed the selection experiment and F.A.S., B.V. and G.A. follow-up experiments. F.A.S., B.V. and S.M. analysed the data. F.A.S., B.V. and S.M. wrote the paper.

## Competing interests

The authors declare no competing interests.
