## [Peer Review File · Nature Communications]

Artificial selection improves pollutant degradation by bacterial communitiesEditorial Note: This manuscript has been previously reviewed at another journal that is not operating a transparent peer review scheme. This document only contains reviewer comments and rebuttal letters for versions considered at *Nature Communications*.

Reviewer #1 (Remarks to the Author):

Dear Editor and Authors,

I have read the authors' responses to my comments. Their thorough responses have resolved my comments regarding the major and minor limitations of the article. I appreciate the effort taken to modify the manuscript and include additional analysis. Thank you.

Reviewer #2 (Remarks to the Author):

The authors have substantially revised their manuscript in response to reviewers, and this reviewer appreciates the large amount of hard work that went into meeting criticisms. I think all criticisms have been addressed as well as can be given the current data. I especially appreciate attempting to come up with posteriori ways to compare selection strategies. I will admit that I am still not fully convinced by the conclusion that this method was better than propagule or migrant pool (the proxies were good, but they were still proxy experiments). BUT, in my mind that is not a problem with the manuscript, just healthy disagreement about the scientific record. At this point I find the manuscript sound, reproducible, and useful to the field.

Reviewer #2 (Remarks on code availability):

Nice clear readme. I have not tested the code but it is in a clean repo and seems well commented.